# An efficient leukemia prediction method using machine learning and deep learning with selected features

Mahwish Ilyas[1], Muhammad Ramzan[2], Mohamed Deriche[3]*, Khalid Mahmood[4], Anam Naz[4]

1 Department of Computer Science, The University of Lahore, Sargodha Campus, Sargodha, Punjab, Pakistan, 2 Department of Software Engineering, University of Sargodha, Sargodha, Punjab, Pakistan, 3 Artificial Intelligence Research Center (AIRC), College of Engineering and Information Technology, Ajman University, Ajman, United Arab Emirates, 4 Department of Information Technology, University of Sargodha, Sargodha, Punjab, Pakistan

* m.deriche@ajman.ac.ae

## Abstract

Leukemia is a serious problem affecting both children and adults, leading to death if left untreated. Leukemia is a kind of blood cancer described by the rapid proliferation of abnormal blood cells. An early, trustworthy, and precise identification of leukemia is important to treating and saving patients' lives. Acute and myelogenous lymphocytic, chronic and myelogenous leukemia are the four kinds of leukemia. Manual inspection of microscopic images is frequently used to identify these malignant growth cells. Leukemia symptoms include fatigue, a lack of enthusiasm, a dull appearance, recurring illnesses, and easy blood loss. Identifying subtypes of leukemia for specialized therapy is one of the hurdles in this area. The suggested work predicts and classifies leukemia subtypes in gene data CuMiDa (GSE9476) using feature selection and ML techniques. The Curated Microarray Database (CuMiDa) collected 64 samples representing five classes of leukemia genes out of 22283 genes. The proposed approach utilizes the 25 most differentiating selected features for classification using machine and deep learning techniques. This study has a classification accuracy of 96.15% using Random Fores, 92.30 using Linear Regression, 96.15% using SVM, and 100% using LSTM. Deep learning methods have been shown to outperform traditional methods in leukemia gene classification by utilizing specific features.

## Introduction

Leukemia, is a kind of cancer caused by abnormality of blood cells that begins with the body creating cancerous white blood cells (WBCs). These aberrant blood cells destroy the blood and bone marrow, which puts the immune system at risk. They can also impair bone marrow's capacity to produce red blood cells and platelets. A full blood count examination is frequently used to identify leukemia. During this test,

**Data availability statement:** The dataset used in this study can be accessed at https://www.kaggle.com/datasets/brunogrisci/leukemia-gene-expression-cumida.

**Funding:** The author(s) received no specific funding for this work.

**Competing interests:** The authors have declared that no competing interests exist.

the doctor will look to see whether the number of WBCs grows and if there are any symptoms of leukemia cells. However, these symptoms are not always sufficient for a clinician to conclude that a patient has leukemia. A bone marrow test is done to confirm the patient's leukemia, followed by a microscopic inspection of a blood smear [1]. All these manual methods of detecting leukemia are entirely dependent on highly qualified medical practitioners and their expertise. Furthermore, manual operations might be time-consuming and costly. Despite progress in cancer classification over the last decade, there is still a need for a fully automated and independent method of cancer diagnosis [2].

Early detection of leukemia is a difficult challenge for professionals nowadays, but it can be performed by utilizing computer-aided automated diagnosis systems in healthcare domains. Machine learning techniques were employed to medical information to create an intelligent detection system. The digital revolution and advances in technological innovation are generating large amounts of data from the medical sectors [3]. Machine learning methodologies are ideal for studying these large data-sets, and numerous techniques have been applied to identify a variety of diseases. Microarray technologies are also causing healthcare facilities to generate a vast number of DNA expression data. Automatic data analysis and categorization are increasingly vital for early diagnosis and making choices in any medical condition. Gene expression data is a popular area of study for machine learning-based biological data processing. Several machine-learning approaches were used to analyze this data [4].

Leukemia is divided into four categories and numerous subgroups. Medical professionals categorise leukemia based on whether leukemia cells originate from myeloid cells and how quickly the disease progresses. Leukemia cells divide rapidly, and the disease advances quickly. If a patient has acute leukemia, he will experience symptoms within weeks of the leukemia cells forming [5]. Acute leukemia is life-threatening and must be treated immediately. Acute leukemia is the most prevalent form of juvenile malignancy [6]. Leukemia is a persistent condition in which immature and mature blood cells coexist. Some cells mature to the point where they can operate as the cells they were designed to be, but not as well as their normal counterparts [7]. In contrast to acute leukemia, the illness typically progresses over time. It might not exhibit signs for years if leukemia is persistent [8]. Biologists can utilize microarray technology to monitor genome-wide properties and at the same time generate gene expression data relevant for cancer classification, which is currently the basis for disease detection. Gene expression data, however, provide difficulties because of their high dimensionality and the presence of redundant, irrelevant, and noisy genes that are not necessary for classification. The classification of gene expression profiles is made more difficult by the use of dimensionality and sparsity concerns. Due to the high number of genes and low number of samples, the curse of dimensionality poses a significant challenge when working with expression data. [9]. If the leukemia facing, one of the growing blood cells becomes uncontrolled, these abnormal cells, also known as leukemia cells, enter the area within the bone marrow. They displace healthy red blood cells, white blood cells, and platelets. The blood has fewer red blood cells, healthy white blood cells, and platelets. Therefore, the organs and

tissues will not receive the oxygen they need to function normally, as well as the body will be unable to combat infections or build blood clots as needed [10].

Existing research has limitations, such as small and non-diverse sample sizes, appropriate feature selection, which limits generalizability, oversimplification of the heterogeneity of leukemia subtypes, limited translation of genomic findings into clinical applications, and technical challenges. To address these difficulties, we proposed machine learning methods with a feasible technique for reducing dimensionality in microarray gene expression research. The scientific contribution of this work is the utilizing of a novel feature selection method to improve the performance of machine learning models for leukemia subtype prediction using high-dimensional gene expression data. We evaluated the proposed technique on the CuMiDa database, which contains gene expression data for five types of leukemia. Machine learning and deep learning differ in leukemia gene classification. ML uses traditional algorithms, while DL uses neural networks. Deep learning improves diagnostic accuracy, early detection, and personalized treatment strategies, improving patient outcomes and quality of life.

Our analysis shows that our proposed methodology is better than existing work by feature selection and classification methods in terms of accuracy, precision, recall and F-measure. The suggested approach has the potential to enhance leukemia diagnosis and therapy by precisely predicting the type of leukemia, resulting in customised therapy plans and better patient outcomes. This work's feature selection and dimensionality reduction algorithms can be applied to complicated biomedical and life sciences datasets.

The remainder of the paper is organized as follows. Section 2 (Related work) describes an analysis of existing studies, Section 3 (Proposed Methodology) discusses the proposed work with their steps, Section 4 (Experimental results and discussion) presents the results obtained using the proposed model, and Section 5 (Conclusion) discusses the conclusion and future work.

## Related work

This section discussed the existing work for leukemia disease detection using gene expression patterns. Explore the key findings, methodologies, feature selection methods and implications of the previous work of researchers. The researchers analyzed gene expression data using a range of approaches, considering current breakthroughs in artificial intelligence (AI) and machine learning. These machine-learning techniques are primarily designed to improve model performance by increasing accuracy and reducing errors.

In this research work, Mahwish Ilyas et. al [11] proposed the leukemia data from the CuMiDa, suggesting work predicts and classifies leukemia subtypes in CuMiDa for categorization, this study used a linear programming approach. The chosen 25 features from the supplied dataset of 22283 features before applying the LP model. These 25 distinctive characteristics were the most important for categorization. This study has a classifying accuracy of 98.44%. This study [12] used microarray gene expression profiles to classify acute myeloid and acute lymphoblastic leukemia using machine learning-based approaches. There were several classifiers used, such as logistic regression, highly random trees, ridge classifier, AdaBoost, linear discriminant analysis, RF, XGB and KNN including PCA, to reduce the dimensions. The study used two different cross-validation techniques to guarantee precise skill assessment. These strategies were assessed using six various performance metrics. Using an eight-fold cross-validation strategy, logistic regression obtained a remarkable classification accuracy of above 99%. State-of-the-art approaches were compared with proposed methodology, and the results showed greater accuracy and shorter computation times.

Recently Nazari et al. [13] came up with a technique in which preprocessed leukemia data was performed by normalization and PCA in Python. Further, the DNN is built for the data and applied to the data where its results are cross validated with classifiers having accuracies of 63.33 and 96.67 using single layer and deep network respectively. In the work of Shafique et al. [14], they developed a deep convolutional neural network to identify the presence of acute lymphoblastic leukemia and categorizing them into four categories using four classes. Fine tuning was made with the use of a pre

 

trained AlexNet. Detection of acute lymphoblastic leukemia performed with 100% sensitivity, 98.11% specificity, 99.50% accuracy; acute lymphoblastic leukemia subtype classification achieved 95.35% sensitivity, 99.4% specificity, and 96.6% accuracy.

Ensaf et al. [15] used the dataset for accurate identification and classification of WBCs as one of 4 class in a scaled down blood cell detection dataset. Here, the proposed model is deep models to feature retrieval and traditional ML classifiers. This uses MobileNet-224 with logistic regression, which achieved 97.03% and the Hybrid model outperforms the baseline FCN by 25.78%. Bozkurt F et al. [16] also presented another approach which included the use of a DCNN to class the blood cells from photographs of blood cells with an accuracy rating of 94% and this implies that a large amount of data can be processed. In the next, Parveen et al. [17] considered the problem of categorization of WBCs identified by them and then the YOLO-v3 object identification methods to be used identifying and locating WBC's by using a bounding box. Then, the suggested work was subjected to a rigorous examination experimentally and suggested to use 99.2% work with 90% accuracy, and then classified. A new semi-supervised white blood cell classifying technique is derived in Yan et al.[18] that consists of FGIA and a SSTS module. FGIA supplies the BCCD dataset, 75 labelled images per category with an average accuracy of 93.2% proving the high performance in the semi-supervised problem of the WBC image classification.

The authors [19] of this study conducted a detailed examination of gene expression data from CuMiDa pertaining to leukemia. They use a set of classification approaches to examine and categorize different forms of leukemia based on gene expression profiles. Different methodologies including DT, LR, SVM, RF, and KNN aim to enhance the understanding of leukemia by utilizing curated datasets. Raja Rajeswari et al. [20] developed a method to identify leukemia by measuring gene expression levels. The major objective of this study was to reduce the rate of leukemia cancer, as chemotherapy can be costly for many individuals. Simsek et al. [21] suggested an ML approach to classifying forms of leukemia using gene data.

The goal of this study [22] was to determine whether the probability of total remission in teenagers with AML on induction therapy could be accurately predicted using ML algorithms that correlate gene expression patterns to RNA sequence data. The trial utilized statistical analysis to identify genes that were distinctly expressed in CR-achieving individuals than those that did not. The study concluded that rather than attempting out, their investigators preferred a combination of algorithms and feature selection strategies while optimizing the hyperparameters of the classifier. A KNN algorithm with the top 75 genes (K=27) achieved the highest area under the curve of 0.84. The remaining 50 genes also demonstrated good performance with an AUC of 0.81 when unseen data were utilized. Recently, Zaied et al. [23] utilized ensemble classifier methods to group acute leukemia gene expression data. The intended data was the sequencing of a patient\u2019s DNA that allowed for the predicting genes and the degree of risk the patient has for leukemia. The Rotation Forest of Bayesian Networks was included in this framework which was the idea of research team. The proposal system is used for categorizing gene expression data findings that are related to acute leukemia. The approach, referred to as, \u201cAggregating Classification Rules by Learning to Rank\u201d, which is the combination of the above two options, has been proven to overcome this part of the problem. In a dataset with a total range of genes from 0 to 20,000 genomes, combinations of pairwise genes are taken and multiplied to and for in a range of 0–1. The joint objective of research was to find the most suitable model for acute leukemia classification. The present investigation comprised a total of 1000 blood smear samples. LeNet-5 successfully detected 85.3% of images with an accuracy of 96.25%, compared to the pre-trained model called, Alex Net's with an accuracy of 98.58% [24]

The author [25] has presented in this study a deep learning system for immature leukemic blast and normal cell identification. In this system, the goal is proposed for a classifier based on the CNN and OD domain properties as supplied by the Discrete Cosine Transform. In addition, the results of the comprehensive test validate that Leukoma is indeed operating. Much work has been done recently to classify gene expression data. The result was obtained using various machine learning algorithms out of which a few were used. For the sake of time and in the rise of the accuracy rate, scientists also

applied a method based on deep learning to gene expression data. With the exception of a few, all the studies based on deep learning of gene expressions were using microarray images. This study, however, had classified previously extracted data of gene features and produced a boosted result of this data compared to other classification methods. In this work [26], the author used a deep convolutional neural network to detect acute lymphoblastic leukemia and categorized four groups of its subtypes. A pre-trained AlexNet model was used, and data was fine tuned on it. It used the data augmentation approach to prevent over training. For acute lymphoblastic leukemia detection, they achieved 100% sensitivity, 98.11% specificity and 99.50% accuracy, and 96.74% sensitivity, 99.03% specificity and 96.06% accuracy for acute lymphoblastic leukemia subtype categorization.

Rehman et al. [27], proposed a technique for identification of ALL subtypes and sensitive bone marrow from damaged images of bone marrow. Robust segmentation and deep learning techniques were used in the convolutional neural network that was employed. Naive Bayesian, SVM and KNN were used for comparison purposes against the experimental findings. The accuracy of the proposed technique was 97.78% based on experimental results. In the research proposed [28], the use of CNN has been made to identify leukemic cells from healthy blood cells. This precise classifier achieved the highest accuracy, specificity, sensitivity, F 1 score of 95.54%, 95.81, 95.91, and precision of 96 respectively. In a proposed novel CS-based technique for categorizing genomic data, Sampathila et al. [28] applied to leukemia subtyping using gene expression analysis. Four statistical features are used in the study for categorizing significant genes. When tested on 7,129 genes in cross-validation with large imbalances, 97.4% classification accuracy was achieved, and 94.3% with a separate data set.

Some essential particle swarm optimization was introduced in this work for both the training and testing process in classification [29]. Individually, 38 and 34 leukemia samples with same 50 genes were used as training and testing datasets respectively. A process of comparison is set out using the Kmeans clustering algorithm. This was further compared with another set of 200 genes. Finally, the performance of K-means is worse than that of PSO and the stability is reversed.

## Proposed methodology

The proposed methodology is summarized as follows: The proposed methodology utilized a novel approach for predicting leukaemia types using high-dimensional gene expression data from the Curated Microarray Database (CuMiDa). Normalisation and absolute transformation are the data preprocessing steps used to standardise features. The data is divided into two sets: training (60%), and testing (40%). The next step is feature selection, which addresses the dataset's high dimensionality by reducing the 22,283 features to the 25 most differentiating ones. This step increases computational efficiency while retaining classification accuracy. The classification process involved applying deep learning and machine learning techniques, followed by analysing findings using several performance measures. The proposed methodology, with its basic steps that are followed in this research study for the leukemia classification, has been shown in Fig 1.

### Data acquisition

CuMiDa's GSE9476 dataset [30] on leukemia gene expression is now available to the public. Feltes et al. established the CuMiDa resource, which includes 78 cancer Microarray datasets that have been extensively cross-checked against 30,000 Gene Expression Omnibus (GEO) articles. GSE9476's sample comprises 64 genes, whereas its feature set includes 22284 genes. 8 cases of Bone Marrow_CD34, 10 instances of Bone Marrow, 26 cases of AML, 10 cases of PB, and 10 cases of PBSC_CD 34 are among the total 64 leukemia samples. It is divided into five groups: AML, PB, PBSC_CD34, Bone Marrow_CD34, and Bone Marrow.

### Data pre-processing

Pre processing is needed to eliminate noise from the data set for the study to be carried out. The initial data was used and the absolute transformation was applied [2]. Changed the data set into a zero mean and a zero standard deviation. Dataset GSE9476 of leukemia gene expression, CuMiDa was imported. It contains 64 samples that represent a different type

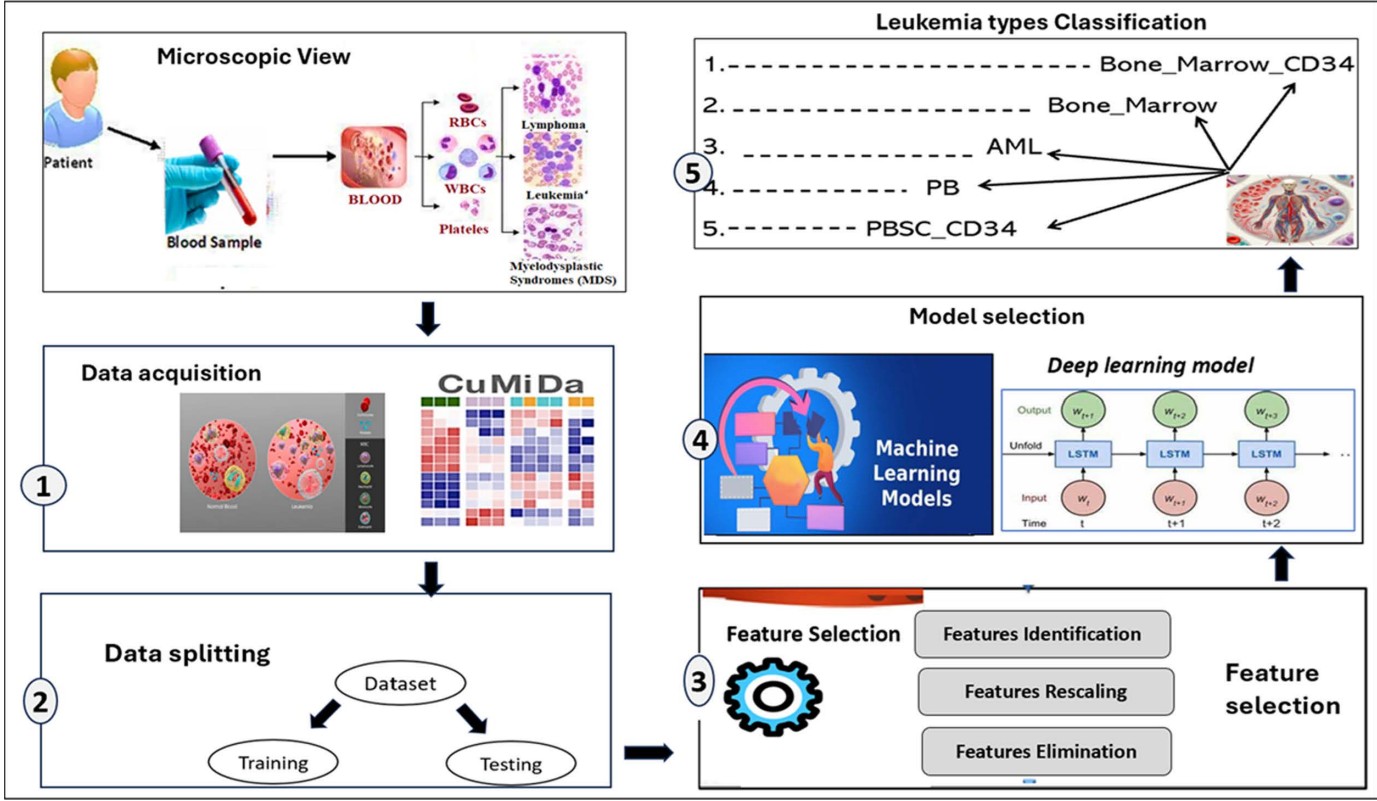

**Fig 1. Basic steps for the proposed methodology for the leukemia classification.**

of leukemia from one of five classes. Here mapping was done of Class_1 (Bone_Marrow_CD34), Class_2 (Bone_Marrow), Class-3 (AML), Class-4 (PB), and Class-5 (PBSC_CD34).

## Features extraction

Given that microarrays can simultaneously measure the expression levels of thousands of genes, they have the potential to accelerate the discovery of new biological knowledge significantly. The second characteristic of microarray data is that the number of variables (genes) 'P' is really much larger than the number of samples ('N'). The sample numbers are N = 64, and the genes are P = 22283 overall, just like in the dataset. To reduce the size of the feature set, we therefore first have to find an acceptable gene selection approach for selected microarray dataset. On the leukemia dataset there are a total of 25 data characteristics. These are the most defining characteristics that will aid in the categorising. The literature presents many methods to attempt to improve classification results and reduce processing costs in gene identification, but most of the proposed techniques are heavy in feature extraction. Therefore, we deleted the elements with higher data separability. Lowering of the features improved the speed and accuracy of the prediction. The ranking of features based on how much they contribute to the classification challenge should be done since it is important to pick a subset of characteristics that improves the categorization. Feature extraction is the name of the process in consideration.

## Features selection algorithm

In the space of biomedical research, particularly in the analysis of microarray data, the challenge lies in the high dimensionality of data where the number of variables G exceeds the sample size N. Our study dataset comprises N = 64

samples and G = 22283 gene variables. To address this, an effective feature selection strategy is imperative to reduce the feature set size, increasing both computational efficiency and classification accuracy.

The algorithm employed for feature selection and rescaling in our study is designed to maximize class reliability, thereby improving the classification performance. The Process is designed the following algorithm is employed for feature rescaling in the form of matrix representation as in equation 1.

$$D = \begin{bmatrix} X_{12} X_{12} \ldots X_{1m} \\ \vdots & \vdots & \vdots & \vdots \\ X_{n1} X_{n2} \ldots X_{nm} \end{bmatrix}$$

(1)

Where the G expression data is represented as matrix D of dimension $\mathbb{R}^{n \times m}$, where N is the sample size and M is the feature selection of genes.

For each feature X, we extract its value across all samples and classes to find a measure for class separability for each feature. We describe the procedure as follows in Equation 2:

$$Let \quad X = [X_{1i} \ X_{2i} \ldots \ldots \ldots X_{ni}]$$

(2)

Where X carries the i$^{th}$ feature value for all the samples from all the classes without loss of generality, we represent X as in equation 3:

$$Let \quad X = [a_1 \ a_2 \ldots \ldots \ldots a_n]$$

(3)

Such that for classwise separation, we classify the samples into c classes, where nj (j = 1, 2, ..., c) is the number of samples in j$^{th}$ classes, as in equation 4.

$$a_1, \ldots \ldots a_{n1} \in C_1$$

$$a_{n+1}, \ldots \ldots a_{n2} \in C_2$$

$$a_{n_{c-1}+1} \ldots \ldots, \ a_{n_c} \in Cc$$

Generally,

$$a_{n_{j-1}+1} \ldots \ldots, \ a_{n_j} \in Cj$$

(4)

The feature value for each class $Cj$ is denoted as $\{a_k\}_{k=1}^{nj} \in Cj$.

We compute the class mean for the feature i-th class names as, in equation 5:

$$m_j = \frac{1}{nj} \sum_{a_{k \in C_j}} a_k$$

(5)

for j = 1,2,.....,c

The deviation of each feature value with the help of equation 6 computed from its class mean is calculated to zero the mean for each class:

$$\{b_j = |a_k - m_j| \ \forall \ a_k \in c_j$$

(6)

Where |.|defines absolute norm. We rewrite equation 2 as:

$$X = [b_1, \ b_2 \qquad b_n]$$ (7)

We compute max separability within class as in equation 8.

$$dk = \bigvee_{b_j \in c_k} b_j$$ (8)

And min separability among the classes is computed as in equation 9:

$$di = \bigwedge_{k=1}^{c} dk$$ (9)

for (i=1,2,......,c)

Where $d_i$ measures the least separability measures among the classes.

Once $d_i$ for (i=1,2,......,c) are computed for each class, we sort features with respect to $d_i$ in descending order. Therefore, we select the best features (as many as possible) suitable for the purpose of classification.

The features are ranked based on their separability measure $di$, and those with higher separability are prioritized. The top 25 features are selected for further classification. This subset was used in conjunction with various supervised machine-learning algorithms to predict leukemia subtypes. The following have been selected 5436, 1756, 9888, 2784, 8673, 6863, 5699, 5294, 3831, 10839, 7616, 9537,188,12159, 2899, 1951, 18418, 7538, 4881, 17796, 15603, 4048, 2539, 1712, 17618.

## Classification techniques

In this study, after the selection of the 25 features using the proposed selection method, the following classifiers are used: Random Forest (RF), Linear Regression (LR), Decision Tree (DT), Support Vector Machine (SVM) with PCA, and LSTM.

Machine learning-based technique for classification.

**Random Forest (RF).** It is a machine learning ensemble learning algorithm. By merging numerous decision trees, It generates a strong prediction model. Each tree is constructed using a random portion of the training data and a random selection of features, which reduces overfitting and improves generalisation. The information pertains to regressions and classifications using random forests (RF) which are known for their high discriminatory accuracy and noise and anomaly tolerance. It employs several different techniques such as bagging and feature randomization. Moreover, it gathered the anticipated results that coincide with tree construction by triggering a majority vote for classification and averaging for regression. The RF is a frequently used tool in a variety of applications, and it barely needs to be tuned to a specific problem so that it can give the most informative features. The RF classificatory comprises three structure-based classifiers. This bagging randomization adds an extra layer of complexity. RF divides each node using the best split among all variables, rather than the best split among a sample of predictors randomly picked at that node. The original data set is utilized to assemble a new training data set with substitute, results computed as in equation 10.

$$\gamma = argmax_{\gamma \in \Upsilon} \sum_{m=1}^{M} I\left(\sum_{i \in N_m} \left[ I\left(x_{fi} \leq \theta_i\right).h_{i,m}^L + I\left(x_{fi} > \theta_i\right).h_{i,m}^R\right] = \gamma\right)$$ (10)

If the condition is true, the indicator function $I$ returns 1, and if it is false, it returns 0. $N_m$ is the set of nodes in tree. $h_{i,m}^L$ and $h_{i,m}^R$ are the predictions for the left and right child nodes i in the tree.

**Decision Tree (DT).** A decision tree is a hierarchical structure of nodes to make predictions or decisions based on input information. It recursively separates the data based on feature values, producing decision nodes representing conditions and leaf nodes representing predictions or outcomes. Each internal node represents a feature, whereas the branches represent the different feature values. Decision trees are understandable and can hold both categorical and numerical data. They can be applied to classification issues with discrete class labels or regression tasks with continuous values as outcomes. Overfitting can arise in decision trees, however, tactics like pruning and ensemble approaches can help to reduce this problem with the help of equation 11.

$$\omega \;=\; T(x) \;=\; \sum_{n \in N} \left[ F(x \in L(n)) \cdot \sum_{c \in C} F\left(c \;=\; argmax_{c \in C} \frac{|\mathbb{Q}_n^c|}{\mathbb{Q}_n}\right) \;+\; F(x \in R(n)) \cdot \sum_{c \in C} F\left(c \;=\; argmax_{c \in C} \frac{|\mathbb{Q}_n^c|}{\mathbb{Q}_n}\right) \right] \tag{11}$$

Where n is the set of all nodes in the tree, $L(n)$ and $R(n)$ are the samples in the left and right child nodes of node n, respectively. $\mathbb{Q}_n^c$ is the proportion of class c in the subset $\mathbb{Q}_n$ over training dataset $\mathbb{Q}$. F is the indicator function that determines the path of traversal in the tree.

**Linear Regression (LR).** Linear regression is a statistical approach for modelling the linear connection between one or more independent variables and a dependent variable. It finds the best-fitting straight line (or hyperplane in multidimensional space) to predict the dependent variable from the independent factors, as computed with the help of equation 12. LR calculates coefficients to minimise the sum of squared differences between projected and actual values, and it is commonly used for forecasting, trend analysis, and analysing variable connections. It is a fundamental approach in statistics as well as machine learning.

$$\delta_{multi} \;=\; argmax_{c \in C} \left(X\beta_c + \beta_{0,c}\right) \tag{12}$$

Where, X is the feature matrix with a sample corresponding to n and p feature, computed as $X \in \mathbb{R}^{n \times \rho}$. $\delta_{multi}$ is the predicted output by selecting the class with the highest value for multiclassification for each class $c \in C$, $\beta$ is the vector coefficient along with $\beta_0$ intercept term with a set of class labels C.

**Support Vector Machine (SVM) with PCA.** SVM is a powerful machine learning algorithm that may be applied to both classification and regression problems. SVM determines the best hyperplane for classifying data or forecasting continuous values while maximizing the margin between classes. It is well-known for its ability to handle high-dimensional data and work efficiently with non-linearly separable data via kernel techniques, as shown in equation 13. SVMs are used in a wide range of fields, including image and text classification, as well as bioinformatics.

$$\hat{y} \;=\; sign\left(\sum_{i=1}^{n} \alpha_i y_i K(xW, \dot{x}_i) + b\right) \tag{13}$$

Where, $\dot{x} = xW$ is the PCA-transformed input value. W is the matrix of top k, principal components with kernel function K. b is the bias term, $\alpha_i y_i$ are the language multipliers obtained from solving the dual optimization problem.

**Deep learning LSTM based model for the classification of the leukemia gene.** The neural network architecture that handles and remembers a sequential input is called Long Short Term Memory (LSTM). This was created in order to overcome standard RNN's vanishing gradient problem. LSTM networks include memory cells, forget gates, input gates, and output gates that allow them to control information flow across the network and store data across long sequences via the softmax function [31]. It is a subclass of recurrent neural networks that excels in sequence prediction tasks and can learn long-term dependencies, computed as in equation 14. LSTM features feedback connections, implying that it can handle individual data points like images and the complete data sequence.

$$\widetilde{p} \;=\; softmax\left(W_p\left(O_T \bigodot \tanh\left(f_T \odot C_{T-1} + i_T \odot \tanh\left(W_{C^{xT}} + U_{C^{h_{T-1}}} + b_c\right)\right)\right) + b_p\right) \tag{14}$$

Where, $\widetilde{p}$ predicted output value, $i_T$ *and* $O_T$ input and output gate vector at time step T respectively. $C_{T-1}$ is the updated cell state over the memory cell. $W_{C^{xT}}$ and $U_{C^{h_{T-1}}}$ are the weight metrics for the gates and connections respectively. $b_c$ and $b_p$ is the bias vector.

## Experimental results and discussion

In this section, the results obtained through the proposed methods are examined and explained. The efficient use of the method is assessed by comparing performance metrics.

### Performance evaluation measures

This paper uses four generally used evaluation metrics [27,32]: Accuracy, Precision, Recall, and F1-Score were stated in Equations 15–18. Specifically, it is the ratio of how many we predict correctly as compared to all instances. The measure of precision shows how much of the predictions of the model are true positive divided by all the predictions of the model turn out to be true positive and avoid false positive. Specifically, recall, or sensitivity, is the ratio of true positive found by the model to the total number of actual positive, so it reflects the model's capacity to recover relevant examples. At the same time, the F measure is a harmonic mean of precision and recall and thus balances the trade-off between the two cases where the dataset is imbalanced. Table 1 is used to calculate the performance measures using the following equation.

### Analysis of the results using deep learning and machine learning

Evaluating model performance, measuring accuracy, and deriving significant insights are all part of the deep learning and machine learning analysis process. It entails assessing feature relevance, detecting patterns or trends in the data, and measuring metrics like accuracy, precision, recall, and F1 score. In order to enhance predictive capacities and accomplish desired results, this analysis aids in model optimisation, parameter optimization, and data-driven decision-making.

### Analysis of results using linear regression

It involves analyzing the results of a linear regression, which involves measuring fit of the model and the importance of predictor variables. The accuracy rate of this analysis is 92% based on the model's performance. Linear regression with 25 selected features is used in Table 2 to display the results. Explanation of each model given in the table has further expanded results in order to expand a better review of the effectiveness and implications of each model. A confusion matrix was used to analyse the results which were Accuracy, Precision, Recall F1-score, as shown in Fig 2.

### Analysis of results using random forest

The model shows good prediction ability with excellent accuracy of 92% same with case of linear regression model shows the generalisation abilities, according to the Random Forest results. According to feature importance analysis, some variables influence predictions more than others, providing important information about the main factors influencing the target variable as shown in Table 3. By using a Random Forest, the Confusion Matrix has been shown in Fig 3.

**Table 1. Performance evaluation measures.**

| Metrics | Formula | |
|---|---|---|
| Accuracy | $\frac{TP+TN}{TP+TN+FP+FN}$ | (15) |
| Precision | $\frac{TN}{TN+FP}$ | (16) |
| Recall | $\frac{TP}{TP+FN}$ | (17) |
| F1-Score | $\frac{2*Precision*Recall}{Preciosion+Recall}$ | (18) |

**Table 2. Performance measure analysis using Linear Regression.**

| Performance measure | Results |
|---|---|
| Accuracy | 0.9230 |
| Precision | 0.9391 |
| Recall | 0.9230 |
| F1 Score | 0.9240 |

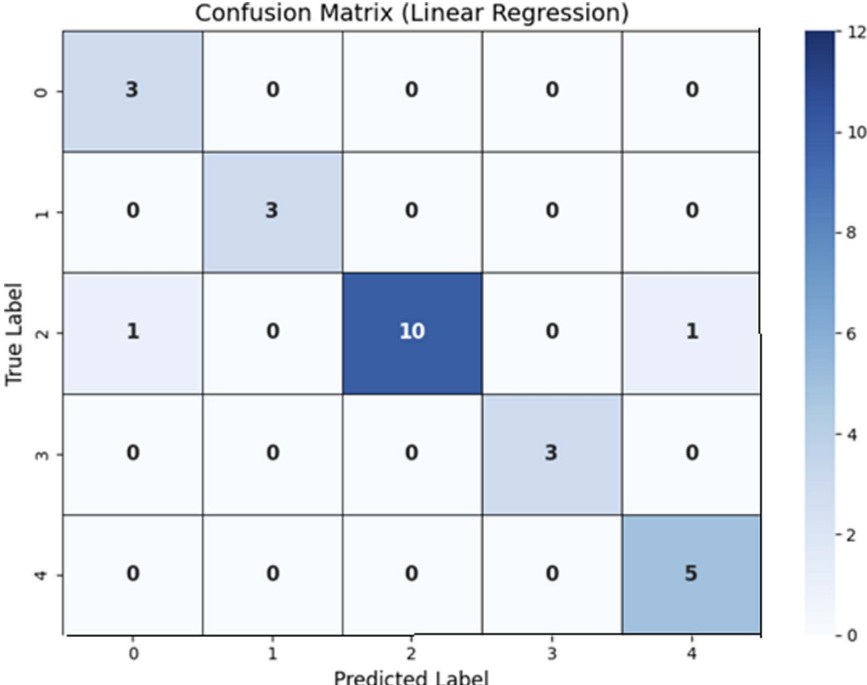

**Fig 2. Confusion Matrix using Linear Regression.**

## Analysis of Results using Support vector machines (SVM)

SVM evaluation of leukemia gene expression data demonstrates high prediction potential for differentiating leukemia subtypes. SVM accomplishes high classification accuracy and precision. Table 4 exhibits that efficient feature selection and hyperparameter variation promote the model's better performance. These findings suggest that SVM is beneficial for classifying Leukemia subtypes and can benefit with diagnosis and cure decisions. The Confusion Matrix has been shown in Fig 4 using a Random Forest technique.

## Analysis of Results using LSTM

The use of LSTM to Leukemia gene expression data demonstrates its efficiency in modelling sequential patterns within gene data. LSTM has a high predictive value, reflecting temporal dependencies that are important in disease subtype classification. Proper hyperparameter tuning and architecture design improve the model's performance, demonstrating its ability to learn and represent complicated patterns., as shown in Table 5 These findings suggest that using LSTM in Leukemia gene expression analysis could lead to more accurate disease diagnosis and treatment strategies.

**Table 3. Results analysis using Random Forest.**

| Performance measure | Results |
|---|---|
| Accuracy | 0.9231 |
| Precision | 0.9391 |
| Recall | 0.9230 |
| F1 Score | 0.9240 |

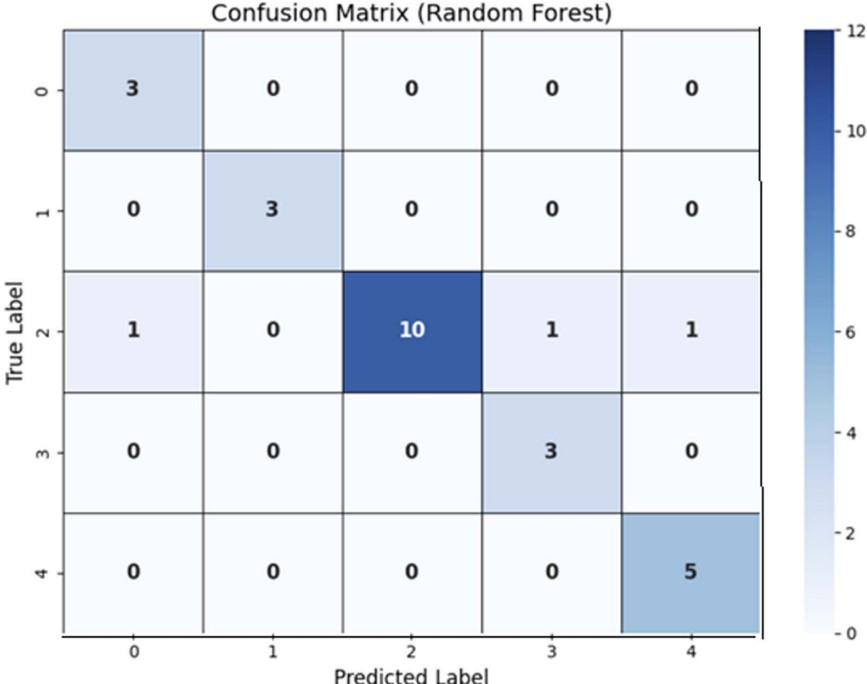

**Fig 3. Confusion Matrix using Random Forest.**

**Table 4. Results analysis using SVM.**

| Performance measure | Results |
|---|---|
| Accuracy | 0.9615 |
| Precision | 0.9711 |
| Recall | 0.9615 |
| F1 Score | 0.9634 |

By using LSTM, the Confusion Matrix has been shown in Fig 5.

The value of the model loss and accuracy using the LSTM model is shown in Fig 6 (a) and (b).

## Comparisons of Results by using applied models

This study presents a comprehensive analysis of three different machine learning models compared with the state-of-the-art deep learning model LSTM trained over 100 epochs evaluated on a dataset with 25 selected features. The models were assessed based on four key metrics. In terms of accuracy, both random forest and linear regression models were

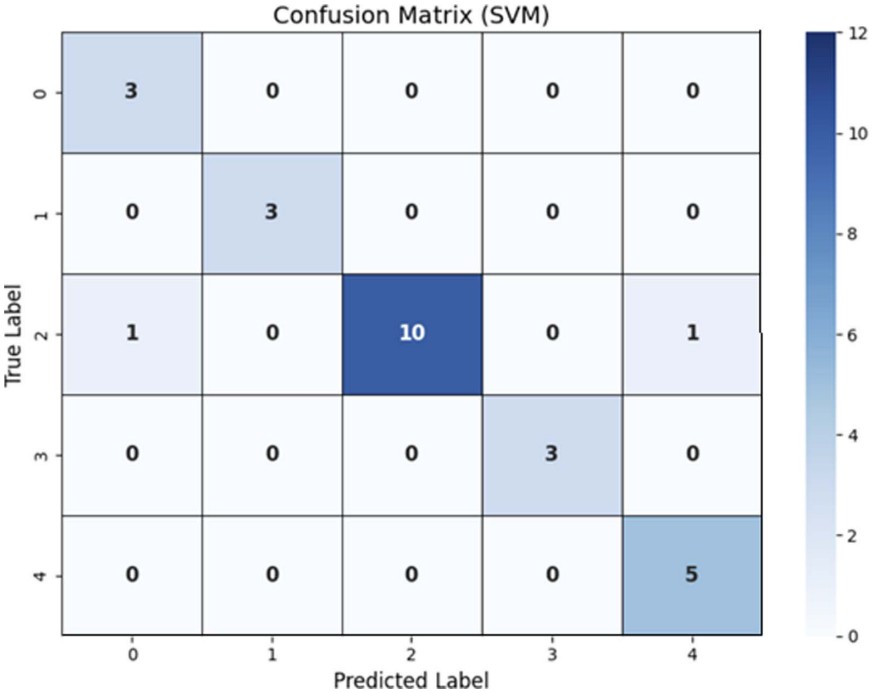

**Fig 4. Confusion Matrix using SVM.**

**Table 5. Results analysis using LSTM.**

| Performance measure | Results |
|---|---|
| Accuracy | 1.0 |
| Precision | 1.0 |
| Recall | 1.0 |
| F1 Score | 1.0 |

able to correctly classify 92% of the instances in the dataset. While among machine learning models, SVM achieves results at 96% over the dataset. Remarkably, the LSTM model being trained with 100 epochs achieved the highest accuracy of 100%. This shows the LSTM model outperformed the other models across all metrics, achieving a perfect score. This suggests that LSTM, with sufficient training, can effectively capture complex patterns in the data, leading to superior performance in leukemia detection. LR, RF and SVM also demonstrated strong performance, making them viable alternatives in terms of accuracy and precision. The results have been discussed in Table 6 using different machine and Deep learning models.

Table 7 presents a comparison for the detection of leukemia prediction. The methods span from 2019 to 2024, showcasing a range of approaches and the number of features utilized with varying accuracy regarding applied models and feature selection. In this regard, our proposed methodology leverages the LSTM network with 25 features, achieving a perfect accuracy of 100%. This indicates that deep learning models particularly LSTM, are highly effective for this application, especially when combined with an optimal selection of features. The progressive improvement in results over the years highlights advancements in both feature selection techniques and applied models. The comparison underscores the potential of LSTM in outperforming traditional ML and Ensemble methods in the context of leukemia prediction. The

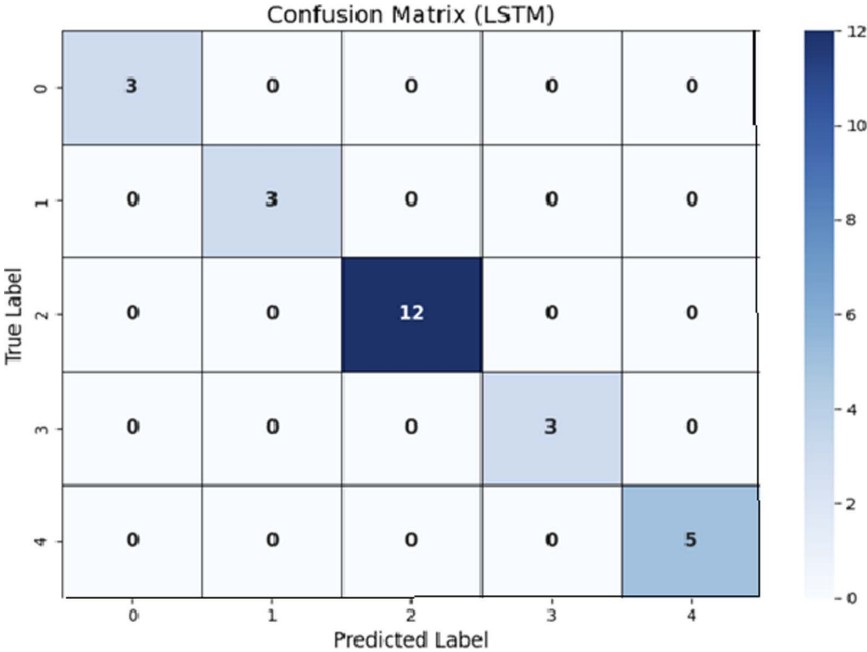

**Fig 5. Confusion Matrix using LSTM.**

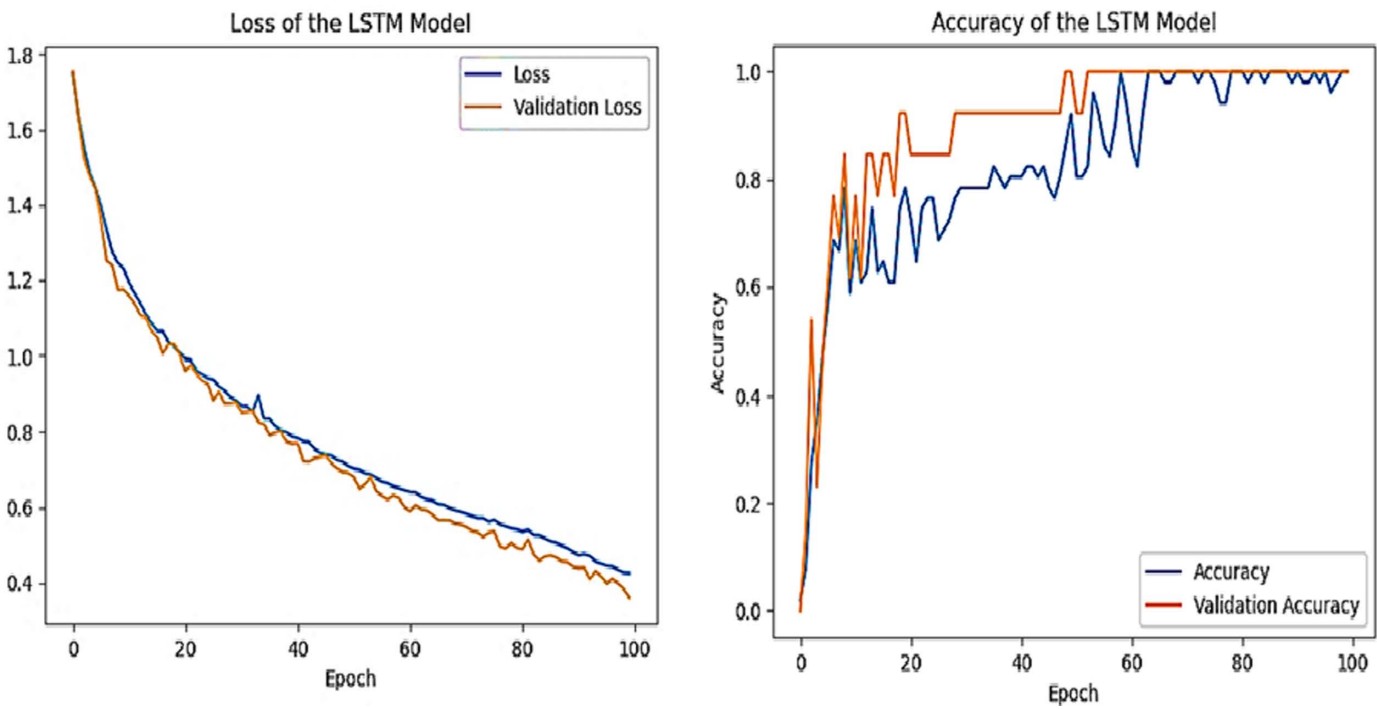

**Fig 6.** (a) Model Loss and (b) Accuracy using LSTM Mode.l.

**Table 6. Comparisons of the Results using Machine Learning and Deep learning models.**

| Model | Accuracy | Precision | Recall | F1 Score | Accuracy 5-Fold Validation |
|---|---|---|---|---|---|
| Random Forest | 0.9231 | 0.9391 | 0.9230 | 0.9240 | 0.9614 |
| Linear Regression | 0.9230 | 0.9391 | 0.9230 | 0.9240 | 0.8964 |
| SVM | 0.9615 | 0.9664 | 0.9615 | 0.9606 | 0.96 |
| LSTM (100 epochs) | 1.0 | 1.0 | 1.0 | 1.0 | 1.0 |

**Table 7. A comparison of the results of the 25 features that were used for this study with other feature sets from earlier leukemia classification studies.**

| Ref | Year | Models | No. of Features | Accuracy |
|---|---|---|---|---|
| [30] | 2019 | SVM | 10 | 0.94 |
| [33] | 2021 | Ensemble models (RF, GB, KNN) | 9 | 0.94 |
| [34] | 2022 | SVM | 16 | 0.90 |
| [11] | 2023 | LP | 25 | 0.98 |
| [35] | 2024 | FLD | 25 | 0.96 |
| **Our Proposed** | | **LSTM**<br>SVM<br>Random Forest<br>Linear Regression | 25 | 1.0<br>0.96<br>0.92<br>0.92 |

accuracy improvements are primarily due to the feature selection and dimensionality reduction method, which extracts the most discriminative features from a high-dimensional dataset while reducing noise and irrelevant variability. The comparisons of the results using machine learning on selected 25 Features and deep learning have been shown in Fig 7.

## Conclusion

One of the most difficult medical challenges is determining which drugs to use for various types of leukemia patients. Advancements in classification models are crucial for improving cancer treatment. In this work, we have used

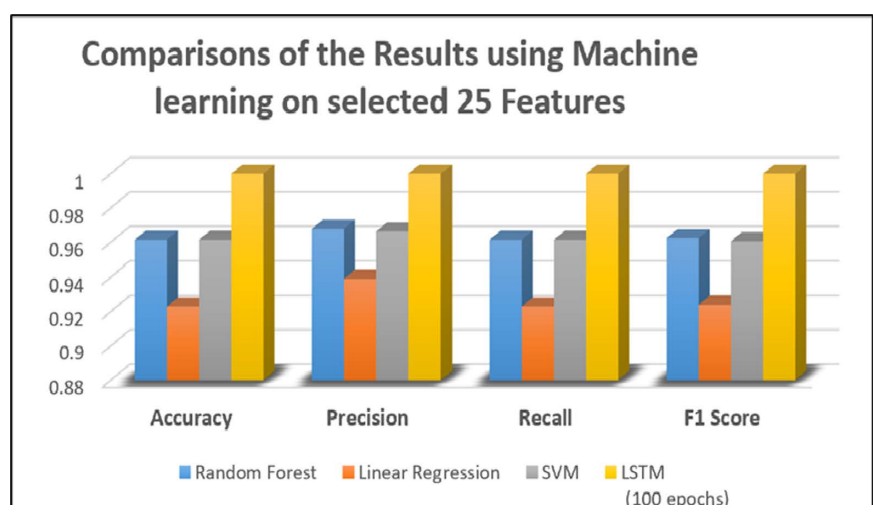

**Fig 7. Comparisons of the Results using Machine learning on selected 25 Features.**

computational machine learning models to identify leukemia subtypes, i.e., Bone Marrow CD34, AML, PB, PBSC CD34. First, we scaled the features of the dataset (22283) and then selected the eminent feature using a state-of-the-art feature selection. 25 different selected relevant traits with high discrimination power have been selected. These features are condensed down to the twenty-five most important in order to speed up the diagnosis. Therefore, the performance of the implemented feature selection method in this work is compared to existing methods on the CuMiDa and it is concluded that the latter may improve the accuracy of machine learning models for high dimensional gene expression datasets.

This work, offers a learning system and feature selection scheme for discovering kinds of leukemia based on gene expression data. For this study, leukemia gene expression data from the Curated Microarray Database were utilized. All the prediction performance was augmented by a rigorous feature extraction and selection technique. The results were analyzed with three different computational models. A new feature selection strategy on mutual information based on Fisher's linear discriminant, linear programming and deep learning's computational models for classification was presented in this study. Many cutting-edge feature selection techniques were compared to the three proposed strategies. It is found that the proposed method achieves better feature selection in identifying the most informative features in classification than several existing methods. Based on the fact that the proposed method is based on information sharing, the interdependence of features is considered rather than features individually, we identified this to be so. A Fisher's Linear Discriminant module was used to create an efficient prediction index that has good consistency first. Overall performance of the proposed framework was excellent with prediction accuracy of 96.92% obtained in less than 100 msec of execution time. The set of features has been used on Second Linear Programming models. This work increased the dataset's accuracy by 98% and played a vital influence in the classification of leukemia subtypes. Our model's overall performance was excellent. The results were analyzed using the third linear regression, random forest, SVM, and LSTM algorithms.

This research contributed to the discovery that when leukemia subtypes are accurately defined and data is fitted with high classification accuracy, cure rates improve while unnecessary toxicities decrease. Because the patient will be able to take preventative measures, doctors will be able to detect the condition earlier. Because clinicians struggle to deal with genetic data and gene expression levels when diagnosing illnesses, our research aids in the early and accurate detection of diseases.

## Supporting information

**S1 Code. Workflow.**
(RAR)

## Acknowledgments

The authors acknowledge the support of the Deanship of Research and Graduate Studies at Ajman University under Projects 2024-IDG-ENIT-1, 2024-IRG-ENIT-36, and 2024-IRG-ENIT-29.

## Author contributions

**Conceptualization:** Muhammad Ramzan, Khalid Mahmood, Anam Naz.

**Data curation:** Mahwish Ilyas, Anam Naz.

**Formal analysis:** Mahwish Ilyas, Mohamed Deriche, Khalid Mahmood, Anam Naz.

**Funding acquisition:** Mohamed Deriche.

**Investigation:** Muhammad Ramzan.

**Methodology:** Mahwish Ilyas, Muhammad Ramzan, Anam Naz.

**Project administration:** Mohamed Deriche.

**Resources:** Mohamed Deriche, Khalid Mahmood.

**Software:** Muhammad Ramzan.

**Supervision:** Khalid Mahmood.

**Validation:** Muhammad Ramzan.

**Visualization:** Muhammad Ramzan.

**Writing – original draft:** Mahwish Ilyas.

**Writing – review & editing:** Mohamed Deriche, Khalid Mahmood, Anam Naz.

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
