## [Decision Letter · Decision Letter 0]

28 Oct 2024

Dear Dr. Ilyas,

Thank you for submitting your manuscript to PLOS ONE. After careful consideration, we feel that it has merit but does not fully meet PLOS ONE’s publication criteria as it currently stands. Therefore, we invite you to submit a revised version of the manuscript that addresses the points raised during the review process.

We look forward to receiving your revised manuscript.

Kind regards,

Javed Rashid, PhD

Academic Editor

PLOS ONE

Journal Requirements:

3. Please upload a copy of Figure 1-6, to which you refer in your text on pages 7 and 14-16. If the figure is no longer to be included as part of the submission please remove all reference to it within the text.

Reviewers' comments:

Reviewer's Responses to Questions

**Comments to the Author**

1. Is the manuscript technically sound, and do the data support the conclusions?

Reviewer #1: Yes

Reviewer #2: Yes

Reviewer #3: Partly

2. Has the statistical analysis been performed appropriately and rigorously?

Reviewer #1: Yes

Reviewer #2: No

Reviewer #3: No

3. Have the authors made all data underlying the findings in their manuscript fully available?

Reviewer #1: Yes

Reviewer #2: No

Reviewer #3: Yes

4. Is the manuscript presented in an intelligible fashion and written in standard English?

Reviewer #1: Yes

Reviewer #2: Yes

Reviewer #3: Yes

Reviewer #1: Areas for Improvement:

Detail on Methodology: The description of feature selection and how the 25 most differentiating features were identified could be elaborated. Details on the criteria for feature selection and how these features contribute to the classification accuracy would be beneficial.

Challenges and Limitations: The text briefly mentions the challenge of identifying leukemia subtypes but does not elaborate on other potential challenges or limitations of the proposed approach. Including this information would provide a more comprehensive view of the research.

Comparison with Existing Methods: It would be helpful to compare the proposed approach with existing methods in more detail. How does it improve upon current methods, and what are its advantages or limitations compared to other state-of-the-art techniques?

Clinical Implications: Adding a discussion on the potential clinical implications of this research, such as how it could impact patient treatment and outcomes, would enhance the relevance of the study.

Reviewer #2: There are several areas need improvement for this research paper. Below, I provide my comments, addressing specific issues related to the content, methodology, and presentation.

1. Consider stating explicitly that the CuMiDa dataset includes only 64 samples, as this sample size poses limitations on the generalizability of the results.

2. How the contributions of the your study differentiate from previous studies would strengthen?

3. Please clarify the preprocessing steps in details, particularly the transformation used for feature normalization. How did the authors handle potential class imbalance?

4. There is a lack of justification for the specific choice of the top 25 features. What criteria were used to determine that these 25 features were the most relevant for leukemia classification? A comparative analysis with different feature set sizes would help validate this choice.

5. The authors use Random Forest, Linear Regression, Decision Tree, SVM, and LSTM. Please justify why these particular classifiers were chosen, especially for comparing ML models with a deep learning model. How do the hyperparameters of these models compare? Providing details about hyperparameter tuning would be helpful.

6. The study does not include cross-validation or other techniques to rigorously evaluate model generalizability. Using k-fold cross-validation and reporting averaged metrics would strengthen the reliability of the reported results.

7. Need more critical discussion of the results, especially the exceptionally high accuracy scores.

8. Restructure the methodology section for greater clarity.

Reviewer #3: The models used (Random Forest, Linear Regression, Decision Tree, SVM, and LSTM) are not cutting edge for current gene expression analysis tasks. Here are my suggestions:

1. The study uses only 64 samples for leukemia classification. This small dataset raises concerns about the generalizability and robustness of the proposed models. For deep learning, especially LSTM, a larger dataset is typically required to avoid overfitting and produce reliable results.

2. The feature selection method is described but lacks sufficient details on how it was validated. There’s no indication that the selected 25 features were evaluated across different splits or external datasets to ensure consistency. The paper does not address the potential risk of data leakage during feature selection.

3. The use of LSTM, a sequential model, isn't fully justified in this context, as gene expression data is not necessarily sequential. The choice of this model seems arbitrary and could be replaced by simpler models better suited for the problem.

4. The LSTM model achieves a perfect 100% accuracy, which is implausible without significant overfitting, especially given the limited dataset size. This result raises concerns about model generalizability, suggesting that the model may be overly trained on the dataset specifics rather than being adaptable to new data.

5. The results are primarily presented in tables without thorough discussion, limiting the reader's understanding of each model's performance and implications.

6. Essential figures, including confusion matrices for each model and training/validation loss graphs for the LSTM model, are missing. These omissions hinder a comprehensive understanding of model performance.

7. Model performances are briefly presented but lack detailed evidence to justify the high accuracy claims. Additional metrics, such as training and validation loss curves or confusion matrices for each model, would improve transparency and support the reliability of these results.

8. Table 7 does not clarify which metric (accuracy, F1-score, or another measure) is being compared for each model, limiting the table’s usefulness in evaluating the relative performance of the models.

9. Performance metrics are inconsistently reported, with some values in percentage (e.g., 90%) and others in decimal format (e.g., 0.90). Adopting a uniform reporting style across the paper would improve readability.

**Do you want your identity to be public for this peer review?** For information about this choice, including consent withdrawal, please see our Privacy Policy

Reviewer #1: **Yes: ** Manuscript Number: PONE-D-24-28100

Manuscript Title: An efficient leukemia prediction method using machine learning and deep learning with Selected Features

Review of the Proposed Work on Leukemia Classification

The text provides an overview of a proposed work that focuses on predicting and classifying leukemia subtypes using gene expression data and machine learning techniques. It highlights the importance of early and accurate diagnosis of leukemia, a severe blood cancer characterized by the rapid proliferation of abnormal blood cells. The text appropriately addresses the significance of accurate classification in treating leukemia and mentions the main subtypes: acute lymphocytic, acute myelogenous, chronic lymphocytic, and chronic myelogenous leukemia.

Strengths: Relevance and Importance: The text underscores the critical need for early and precise identification of leukemia to improve patient outcomes. This focus on early detection aligns well with current priorities in cancer research and treatment.

Data and Methodology: The use of the Curated Microarray Database (CuMiDa) with 64 samples and the application of feature selection and machine learning techniques is well-explained. The mention of specific techniques such as Random Forest, Linear Regression, SVM, and LSTM provides a clear view of the methodological approach.

Performance Metrics: The classification accuracy reported (96.15% for Random Forest and SVM, 92.30% for Linear Regression, and 100% for LSTM) is impressive and suggests that deep learning methods, particularly LSTM, outperform traditional methods in this context.

Areas for Improvement:

Detail on Methodology: The description of feature selection and how the 25 most differentiating features were identified could be elaborated. Details on the criteria for feature selection and how these features contribute to the classification accuracy would be beneficial.

Challenges and Limitations: The text briefly mentions the challenge of identifying leukemia subtypes but does not elaborate on other potential challenges or limitations of the proposed approach. Including this information would provide a more comprehensive view of the research.

Comparison with Existing Methods: It would be helpful to compare the proposed approach with existing methods in more detail. How does it improve upon current methods, and what are its advantages or limitations compared to other state-of-the-art techniques?

Clinical Implications: Adding a discussion on the potential clinical implications of this research, such as how it could impact patient treatment and outcomes, would enhance the relevance of the study.

Overall, the proposed work appears to be a promising approach to leukemia classification using gene expression data and advanced machine learning techniques. It demonstrates significant potential in improving diagnostic accuracy and could have substantial implications for personalized treatment strategies. Addressing the suggested improvements would strengthen the proposal and provide a more comprehensive understanding of the research’s impact and methodology.

Reviewer #2: **Yes: ** Muhammad Sohail

Reviewer #3: No

---

## [Author Response · Author response to Decision Letter 1]

23 Dec 2024

All the comments of three reviewers and Journals Comments have been solved and addressed in the updated manuscript.

---

## [Decision Letter · Decision Letter 1]

22 Jan 2025

Dear Dr. Ilyas,

Thank you for submitting your manuscript to PLOS ONE. After careful consideration, we feel that it has merit but does not fully meet PLOS ONE’s publication criteria as it currently stands. Therefore, we invite you to submit a revised version of the manuscript that addresses the points raised during the review process.

We look forward to receiving your revised manuscript.

Kind regards,

Javed Rashid, PhD

Academic Editor

PLOS ONE

Journal Requirements:

Additional Editor Comments:

Please revise the manuscript as per reviewer's comments and resubmit.

Reviewers' comments:

Reviewer's Responses to Questions

**Comments to the Author**

Reviewer #1: All comments have been addressed

Reviewer #2: All comments have been addressed

Reviewer #3: All comments have been addressed

2. Is the manuscript technically sound, and do the data support the conclusions?

Reviewer #1: Yes

Reviewer #2: Yes

Reviewer #3: Yes

3. Has the statistical analysis been performed appropriately and rigorously?

Reviewer #1: Yes

Reviewer #2: Yes

Reviewer #3: Yes

4. Have the authors made all data underlying the findings in their manuscript fully available?

Reviewer #1: Yes

Reviewer #2: Yes

Reviewer #3: Yes

5. Is the manuscript presented in an intelligible fashion and written in standard English?

Reviewer #1: Yes

Reviewer #2: Yes

Reviewer #3: Yes

Reviewer #1: The manuscript presents an important and timely study on the classification of leukemia subtypes using gene expression data and machine learning (ML) techniques. The authors leverage the CuMiDa dataset to predict leukemia subtypes with high accuracy, employing feature selection and both traditional and deep learning methods. The results demonstrate promising outcomes, with the Long Short-Term Memory (LSTM) model achieving an impressive 100% accuracy. This study provides valuable insights into the application of computational techniques for advancing leukemia diagnostics and precision medicine.

The topic is highly relevant and addresses a critical health issue, emphasizing the importance of early and accurate leukemia diagnosis.

The study's aim—to classify leukemia subtypes using gene data and machine learning—is clearly stated, making it easy for the reader to understand the objective.

Suggestions for Improvement:

Add a brief background on how machine learning and deep learning techniques differ in their approach to classifying leukemia genes.

Include a short discussion on the clinical implications of the proposed approach. How might this enhance leukemia treatment and patient outcomes?

Reviewer #2: The revised manuscript has thoroughly addressed all my comments, significantly improving its clarity, structure, and scientific rigor. I am pleased to recommend the manuscript for acceptance.

Reviewer #3: I am satisfied with the revisions made and am pleased to accept it for consideration.

The authors have addressed all the points I raised in my previous review, and the paper now meets the required standards. I believe it will be a valuable contribution to the field.

**Do you want your identity to be public for this peer review?** For information about this choice, including consent withdrawal, please see our Privacy Policy

Reviewer #1: **Yes: ** Biba Vikas

Reviewer #2: No

Reviewer #3: No

---

## [Author Response · Author response to Decision Letter 2]

5 Feb 2025

All comments have been resolved and updated in the manuscript.

---

## [Decision Letter · Decision Letter 2]

23 Feb 2025

An efficient leukemia prediction method using machine learning and deep learning with Selected Features

PONE-D-24-28100R2

Dear Dr. Ilyas,

We’re pleased to inform you that your manuscript has been judged scientifically suitable for publication and will be formally accepted for publication once it meets all outstanding technical requirements.

Kind regards,

Javed Rashid, PhD

Academic Editor

PLOS ONE

Additional Editor Comments (optional):

Reviewers' comments:

Reviewer's Responses to Questions

**Comments to the Author**

Reviewer #1: All comments have been addressed

2. Is the manuscript technically sound, and do the data support the conclusions?

Reviewer #1: Yes

3. Has the statistical analysis been performed appropriately and rigorously?

Reviewer #1: Yes

4. Have the authors made all data underlying the findings in their manuscript fully available?

Reviewer #1: Yes

5. Is the manuscript presented in an intelligible fashion and written in standard English?

Reviewer #1: Yes

Reviewer #1: Thank you for your revised manuscript. The revisions have significantly improved the clarity and scientific rigor of the paper.

**Do you want your identity to be public for this peer review?** For information about this choice, including consent withdrawal, please see our Privacy Policy

Reviewer #1: **Yes: ** Biba Vikas

---

## [Editor Report · Acceptance letter]

PONE-D-24-28100R2

PLOS ONE

Dear Dr. Ilyas,

I'm pleased to inform you that your manuscript has been deemed suitable for publication in PLOS ONE. Congratulations! Your manuscript is now being handed over to our production team.

Kind regards,

on behalf of

Dr. Javed Rashid

Academic Editor

PLOS ONE